# Doing Fast Adaptation Fast: Conditionally Independent Deep Ensembles for Distribution Shifts

## Abstract

Classifiers in a diverse ensemble capture distinct predictive signals, which is valuable for datasets containing multiple strongly predictive signals. Performing fast adaptation at test time allows us to generalize to distributions where certain signals are no longer predictive, or to avoid relying on sensitive or protected attributes. However, ensemble learning is often expensive, even more so when we need to enforce diversity constraints between the high-dimensional representations of the classifiers. Instead, we propose an efficient and fast method for learning ensemble diversity. We minimize conditional mutual information of the output distributions between classifiers, a quantity which can be cheaply and exactly computed from empirical data. The resulting ensemble contains individually strong predictors that are only dependent because they predict the label. We demonstrate the efficacy of our method on shortcut learning tasks. Performing fast adaptation on our ensemble selects shortcut-invariant models that generalize well to test distributions where the shortcuts are uncorrelated with the label.

## 1 Introduction

Some of the strongest scientific theories are supported by multiple sources of evidence, a principle described by 19th century philosopher William Whewell as "consilience". Evolution is one such example, having been firmly corroborated by fields ranging from paleontology to genetics. In many real-world applications of machine learning, datasets can similarly contain multiple predictive signals that explain the label well. In these settings, a standard model typically learns from a combination of predictive features (Ross et al., 2018; Kirichenko et al., 2022). Such a model will fail to generalize to *distribution shifts* that break the correlation between certain signals and the label (Hovy & Søgaard, 2015; Hashimoto et al., 2018; Puli et al., 2022).

This shortcoming can be addressed by learning a *diverse* set or ensemble of classifiers. Such methods typically exploit some notion of independence to learn multiple classifiers that rely on different predictive signals. We can then perform **fast adaptation**, using a small amount of out-of-distribution (OOD) validation data to select the model that generalizes best. Learning diversity is also beneficial in and of itself: these classifiers are empirically shown to be more human-interpretable than if we were to fit a single model (Ross et al., 2018), possibly because they learn disentangled representations that correspond to natural factors of variation (Shu et al., 2019).

The key challenge is quantifying the right notion of diversity. Existing work has exploited concepts like input gradient or parameter orthogonality as a proxy for statistical independence (Teney et al., 2021; Xu et al., 2021). To tackle OOD generalization, which fundamentally requires additional assumptions or data beyond the observed training data (Bareinboim et al., 2022; Schölkopf et al., 2021), previous work have also assumed access to unlabelled test data and measured disagreement on those examples (Lee et al., 2022; Pagliardini et al., 2022). However, these objectives or assumptions are often prohibitive or unrealistic in real-world settings. For example, group-balanced test data is not always obtainable, e.g. when deploying a pneumonia model to multiple new hospitals whose patient profiles may change over time. Another costly example is enforcing input gradient orthogonality on high-dimensional covariates like images or text, where it can be challenging to avoid learning from orthogonal covariates of the same underlying feature, such as neighboring pixels.

To avoid the pitfalls of operating in high-dimensional input or parameter space, a promising line of work instead adopts the *information-theoretic* perspective and tackles the problem as representation learning. These approaches apply the information bottleneck method and minimize mutual information between the representations learnt by each classifier. Such an objective forces the classifiers to rely on distinctly meaningful features for prediction. Most notably, Pace et al. (2020) and Rame & Cord (2021) minimize mutual information between the classifier representations *conditioned on the label*. Since any pair of predictors cannot both be accurate while remaining unconditionally independent, the extra conditioning prevents learning weak classifiers. The resulting ensemble contains accurate classifiers that nevertheless rely on distinct predictive signals. The only core assumption is that the underlying predictive signals are themselves conditionally independent.

These approaches are conceptually appealing but practically challenging. Mutual information between high-dimensional representations is intractable and must be approximated, either via variational (e.g. Fischer, 2020) or contrastive (e.g. Oord et al., 2018) bounds. Furthermore, such approximations are computationally expensive, a problem that is compounded in the ensemble setting where we wish to train *multiple* classifiers speedily.

We seek to learn ensemble diversity fast *and* effectively. Our key insight is that it suffices to enforce conditional independence on the output distributions of the classifiers. Our first contribution is proposing **conditional mutual information (CMI) between output distributions** as the regularizing objective. Assuming conditionally independent predictive signals, enforcing CMI between output distributions also guarantees that the ensemble where separate predictive signals are learnt by separate classifiers is a minimizing solution. Since the output distribution is categorical, CMI can be *cheaply and exactly computed* from empirical data. In addition, our method avoids using additional sources of data that cannot be found in many real-world domains, such as unlabelled test data or "group" labels for each predictive signal in the dataset. We only permit a small amount of validation data from the test distribution for (1) hyperparameter tuning and (2) selecting the final predictor from our ensemble. We dub our approach as **Conditionally Independent Deep Ensembles** (CoDE).

Our second contribution is evaluating CoDE on benchmark datasets for **shortcut learning** (Geirhos et al., 2020). Shortcuts are signals that are **(i)** highly but *spuriously correlated* to the label in the training distribution, possibly due to biases in data collection or other systematic pre-processing errors (Torralba & Efros, 2011), and **(ii)** *preferentially* learnt by a neural network, possibly due to simplicity biases (Shah et al., 2020) or architectural biases (e.g. convolutional neural networks (CNNs) relying on texture over shape (Baker et al., 2018)). An empirical risk minimizing (ERM) model will rely on shortcuts and fail to generalize to test distributions where they are no longer correlated to the label. This is a natural application for our method as the core assumption of conditional independence applies to many such datasets — for example, in natural images, the foreground is typically the label and is thus conditionally independent from the background (shortcut). We show that CoDE effectively recovers an ensemble where the shortcut features and the true signal are learnt by separate classifiers.

## 2 PRELIMINARIES: SETUP AND NOTATION

In Section 3, we will fully motivate the assumptions behind our model of the data-generating process (DGP). However, we describe it here first to establish key terminology and concepts.

**Data-Generating Process** Let $\mathbf{z}$ denote the set of latent factors that generate the set of observed features $\mathbf{x} \in \mathbb{R}^P$. Let $y \in \{0, 1, \dots, K-1\}$ denote the label. The data $p_e(\mathbf{x}, y, \mathbf{z})$ is generated from a family of distributions indexed by $e$, the environment. We only consider: (i) a single training environment ($e = tr$), from which we have access to i.i.d. labelled training examples $D_{tr} = \{\mathbf{x}_i, y_i\}_{i=1}^N$, and (ii) a test environment ($e = te$), from which we draw unlabelled test examples that our model should perform well on. We also allow access to a small set of labelled validation data $D_{val} = \{\mathbf{x}_i, y_i\}_{i=1}^{N'}$ from the test environment, which is used only for hyperparameter tuning and ensembling (i.e. constructing the final model from the set of learnt classifiers).

We make the following assumptions on the DGP:

(i) all label information is encoded by $\mathbf{z}$, i.e. $p_e(y|\mathbf{x}, \mathbf{z}) = p_e(y|\mathbf{z})$ for all $e$

(ii) $p_e(\mathbf{x}|\mathbf{z}) = p(\mathbf{x}|\mathbf{z})$ is invariant across all $e$

(iii) $p_e(\mathbf{z}) > 0$ for all $e$ and $\mathbf{z}$

(iv) $p_e(y) > 0$ for all $e$ and $y$

(v) **[Latent Conditional Independence]** $z_i \perp\!\!\!\perp z_j \,|\, y$ for all $e$ and $i, j$

Based on these assumptions, we can factorize $p_e(\mathbf{x}, y, \mathbf{z})$ as:

$$p_e(\mathbf{z}, \mathbf{x}, y) = p_e(y) \left( \prod_{i=1}^{L} p_e(z_i|y) \right) p(\mathbf{x}|\mathbf{z}) \tag{1}$$

*Example: ColoredMNIST*   As introduced in Arjovsky et al. (2019), $y$ is a binary label which determines color ($z_1 \in \{\text{red}, \text{green}\}$) with probability $p_c$ and digit ($z_2 \in \{\text{0-4}, \text{5-9}\}$) with probability $p_d$. $p_c$ and $p_d$ are independently chosen. In the training distribution, $p_c = 0.25$ and $p_d = 0.1$, as such, an ERM model will primarily learn from color. $p_c$ and $p_d$ can be arbitrary in the test distribution.

*Example: Waterbirds*   As introduced in Sagawa et al. (2019), $y$ is a binary label determining if the image represents a water or land bird. It perfectly determines the foreground ($z_1 \in \{\text{water bird}, \text{land bird}\}$) and is highly but spuriously correlated to the background ($z_2 \in \{\text{water}, \text{land}\}$) in the training distribution. An ERM model will learn from background features.

**Group Robustness**   When $\mathbf{z}$ is discrete, each possible value that $\mathbf{z}$ can take is known as a *group*. Due to the spurious correlations created by $p_{tr}(z_i|y)$, groups that are highly represented in the training set are called "majority groups", and poorly-represented groups are "minority groups". Group robustness refers to the goal of generalizing well on all groups and is one natural way of evaluating if a model has been learning shortcuts. For example, both `ColoredMNIST` and `Waterbirds` admits four groups formed by the Cartesian product of $z_1$ and $z_2$.

**Ensembles and Fast Adaptation**   A classifier $f(\mathbf{x}) := p_\theta(y|\mathbf{x})$ is parametrized by $\theta$ and outputs class probabilities. We will use $\hat{y} := p_\theta(y)$ to denote the unconditional output distribution. We use the term "ensemble" loosely to refer to a set of $M$ classifiers $\{f_m\}_{m=1}^{M}$ that can be learnt jointly or sequentially. (Section 4 clarifies the relationship to traditional ensemble methods.) After all $M$ classifiers are learnt, the final model $\theta^*$ is selected using validation data $D_{val}$:

$$\theta^* = \arg \min_{\theta_m, m \in \{1, \ldots, M\}} \frac{1}{N'} \sum_{i=1}^{N'} \log p_{\theta_m}(y_i|\mathbf{x}_i) \tag{2}$$

This process is referred to as **fast adaptation**.

## 3   CONDITIONALLY INDEPENDENT DEEP ENSEMBLES

To motivate our approach and the assumptions made in (1), we first define what it means to learn a *diverse* ensemble and explain why conditional independence is a sound measure of diversity.

### 3.1   DIVERSITY AS CONDITIONAL INDEPENDENCE

Diverse classifiers utilize separate predictive signals, intuitively, they predict the *"same things for different reasons"* (Rame & Cord, 2021). Our setup in Section 2 formalizes this notion of "different reasons" by explicitly defining the latent variable $\mathbf{z}$, which models the total underlying set of predictive signals that relate $\mathbf{x}$ to $y$. A classifier that learns a mapping from $\mathbf{x}$ to $y$ can then be interpreted as implicitly inferring $\mathbf{z}$ from $\mathbf{x}$ and learning a mapping from $\mathbf{z}$ to $y$. We can thus define diverse classifiers that rely on separate predictive signals as learning from **separate dimensions or subspaces of $\mathbf{z}$**.

To formalize the idea that a classifier $f$ learns using only a subspace of $\mathbf{z}$, one naive approach might be to define $f$ as relying only on the subspace $\mathbf{z}_{[a]}$ if and only if (some distribution computed from) $f$ is independent of its complement $\mathbf{z} \backslash \mathbf{z}_{[a]}$. This definition is convenient as it suggests that the appropriate objective to learn a diverse ensemble is simply to enforce statistical independence between the classifiers. This follows because two classifiers that rely on overlapping subspaces of $\mathbf{z}$ will necessarily be dependent.

However, the definition above assumes that distinct predictive signals (i.e. subspaces of $\mathbf{z}$) are themselves unconditionally independent. This is not always true when a dataset contains multiple strongly predictive signals. Dimensions of $\mathbf{z}$ can be dependent by virtue of their correlation to $y$. Classifiers that learn from such signals will similarly be dependent. Shortcut learning is precisely a problem because meaningful and spurious features are highly correlated in the training environment.

This conundrum can be resolved by establishing independence of the latent factors **with conditioning on** $y$. Doing so is equivalent to assuming that upon knowing the true label, observing one set of features yields no *additional* information about other features. This is usually a realistic assumption to make. As the *Waterbirds* example in Section 2 shows, backgrounds and foregrounds are often conditionally independent in the test distributions we care about. This motivates our assumption **(v)** of *latent conditional independence* in Section 2, where the individual factors $z_i$ are conditionally independent given $y$. We formalize this notion of "diversity as conditional independence" below.

**Definition 3.1.** Let $\mathbf{z}_{[a]} := (z_{a_1}, \ldots, z_{a_l})$ denote some subspace of $\mathbf{z}$. Let $\hat{h}(f)$ denote some distribution computed from $f$. We say $f$ is **invariant** to $\mathbf{z}_{[a]}$ if $\hat{h} \perp\!\!\!\perp (z_{a_1}, \ldots, z_{a_l}) | y$. Let $\mathbf{z}_{[i]}$ be the maximal subset of $\mathbf{z}$ that $f$ is invariant to. Then $f$ is said to **rely on** $\mathbf{z}_{-[i]} := \mathbf{z} \backslash \mathbf{z}_{[i]}$ for prediction.

**Definition 3.2.** Let $f$ and $f'$ be a pair of classifiers that rely on $\mathbf{z}_{[i]}$ and $\mathbf{z}_{[i']}$ respectively. $f$ and $f'$ are said to be **diverse** if $\mathbf{z}_{[i]} \cap \mathbf{z}_{[i']} = \emptyset$. An ensemble $\{f_m\}_{m=1}^{M}$ is **diverse** if every pair of classifiers $f_j, f_k$ in the ensemble are diverse.

It follows immediately from Definition 3.2 that diverse classifiers must themselves be conditionally independent, i.e. $\hat{h}_i \perp\!\!\!\perp \hat{h}_j | y$. Our training objective for learning a diverse ensemble should therefore enforce conditional independence on all pairs of classifiers:

$$\arg \max_{\theta_1, \ldots, \theta_M} \sum_{i=1}^{N} \sum_{m=1}^{M} \log p_{\theta_m}(y_i | \mathbf{x}_i) \tag{3}$$
$$\text{subject to} \quad \hat{h}_s \perp\!\!\!\perp \hat{h}_t \,|\, y \qquad \forall s, t$$

We can interpret (3) as follows: the main objective guarantees that the learnt ensemble contains *individually strong* predictors, whereas the constraint guarantees that each predictor is *uninformative* of the others when conditioned on the label. Put together, (3) learns classifiers that rely on conditionally independent subspaces of $\mathbf{z}$ and thus provide no additional information about each other. As is typical in machine learning (Krogh & Hertz, 1991; Deb, 2014), we optimize an unconstrained analogue of (3) by expressing the constraint as a regularization term.

## 3.2    Enforcing Conditional Independence via Output Distributions

It remains for us to decide on the distribution $\hat{h}$ that we constrain, as well as the (unconstrained) regularization objective from (3). These choices are crucial in many ways. Since independence with respect to $\hat{h}$ underpins the notions of invariance and diversity in Definitions 3.1 and 3.2, it must be informative about the underlying predictive signals that a classifier is relying on. Furthermore, $\hat{h}$ and the regularization objective must be tractable.

Earlier work such as Pace et al. (2020) and Rame & Cord (2021) choose $\hat{h}$ to be the representations learnt by the classifiers, e.g. by constructing $f = f_l \circ f_e$ as a deep encoder network $f_e$ that is attached to a linear classifier $f_l$ and letting $\hat{h} = f_e(\mathbf{x})$. As the regularization objective for conditional independence, Rame & Cord (2021) compute pairwise conditional mutual information $\mathcal{CMI}(f_{e,s}, f_{e,t})$ whereas Pace et al. (2020) compute total correlation $\mathcal{TC}(f_{e,1}, \ldots, f_{e,M})$. Since the encoder representations are high-dimensional, these terms must be approximated.

We propose a far simpler and more efficient method. Instead of network representations, we choose $\hat{h}$ to simply be the output distribution $\hat{h} = f(\mathbf{x}) = p_\theta(y|\mathbf{x})$ of the classifier. Accordingly, our regularization objective is **conditional mutual information (CMI) between the output distributions** of the classifiers. For any pair of classifiers $f_j, f_k$, we have:

$$\mathcal{CMI}(f_s, f_t) = \mathbb{E}_y \left[ \mathcal{D}_{KL}\Big( p(f_s, f_t | y) \,\|\, p(f_s | y) p(f_t | y) \Big) \right] \tag{4}$$

CMI is zero iff $f_s \perp\!\!\!\perp f_t | y$ for all values of $y$. Enforcing conditional independence on the classifiers' predicted output probabilities rather than underlying representations trades off granularity of the independence constraint for computational efficiency. We believe that this is a valuable trade-off. Since $\hat{y}$ has categorical support, (4) can be cheaply and exactly estimated from training data. As our experiments in Section 5 show, even on a noisier signal like output distributions, enforcing conditional independence is sufficient to learn a diverse ensemble.

Even though a diverse ensemble implies pairwise conditionally independent classifiers, the converse is not necessarily true. Mutual information is also zero if one of the classifiers outputs random or constant class probabilities. In particular, optimizing a weighted sum of the cross-entropy term and the CMI term can be challenging — overly weak regularization produces an ensemble that is not diverse, whereas overly strong regularization tends towards solutions containing close-to-random classifiers. Instead, we propose adding another term to regularize for confident predictions:

$$\mathcal{R}(f) = \sum_{k=1}^{K} \left\| p(\hat{y}|y=k) - I_k \right\| \tag{5}$$

where $I_k$ is the indicator function at $k$. Put together, the **overall loss objective** is:

$$\mathcal{L}(\{\theta_m\}_{m=1}^{M}) = \sum_{i=1}^{N} \sum_{m=1}^{M} \log p_{\theta_m}(y_i|\mathbf{x}_i) + \lambda_1 \cdot \sum_{s=1}^{M} \sum_{t=1}^{s-1} \mathcal{CMI}(f_s, f_t) + \lambda_2 \cdot \sum_{m=1}^{M} \mathcal{R}(f_m) \tag{6}$$

where $\lambda_1$ and $\lambda_2$ are hyperparameters controlling the strength of regularization. A solution that minimizes (6) contains an ensemble where: **(i)** each classifier is accurate *(first term)* and confident *(third term)*, and **(ii)** different classifiers rely on different subspaces of $\mathbf{z}$ for prediction *(second term)*. We name such an ensemble a **Conditionally Independent Deep Ensemble** (CoDE).

### 3.3 CODE: COMPUTATIONAL DETAILS

The hyperparameters of the method are $M$, $\lambda_1$, and $\lambda_2$. Unlike traditional ensembles, $M$ (ensemble size) will typically be small ($M = 2$ for all our experiments) since $M$ cannot be larger than the number of conditionally independent predictive signals inherent in the dataset. As is typical for OOD problems, we assume access to validation data from the test environment for hyperparameter tuning.

Objective (6) describes the situation where all $M$ classifiers are *jointly* optimized. Since $M$ is typically small, doing so is not difficult or computationally expensive (as might be with traditional ensembles). An alternative to joint optimization is to learn the classifiers in a *sequential* fashion. The analogue to (6) becomes:

$$\mathcal{L}(\theta_m) = \sum_{i=1}^{N} \log p_{\theta_m}(y_i|\mathbf{x}_i) + \lambda_1 \cdot \sum_{s=1}^{m-1} \mathcal{CMI}(\hat{y}_s, \hat{y}_m) + \lambda_2 \cdot \mathcal{R}(f_m) \tag{7}$$

Sequential optimization presents a natural way to determine $M$, as we can terminate the training process when no more predictive classifiers can be learnt. However, it will fail if earlier classifiers in the sequence learn multiple predictive signals. We discuss this further in Section 5.

## 4 RELATED WORK

**Ensemble Methods**    In statistics, ensembling traditionally refers to combining multiple predictors into a single model that outperforms the individual learners, typically by bagging (Breiman, 1996) or boosting (Schapire, 1990). Diversity in this context refers to minimizing correlation between individual learners, which reduces variance and improve generalization (Kuncheva & Whitaker, 2003). Deep ensembling (Lakshminarayanan et al., 2017) is an analogous approach in deep learning where multiple randomly-initialized networks are trained in parallel, however, they are generally used for the purpose of uncertainty estimation. Unlike these works, we consider diversity specifically in the context of datasets with multiple predictive signals, and learning a diverse ensemble as recovering all such signals for the purpose of OOD generalization.

**Various Approaches For Learning Diversity**    As an unsupervised task, diversity refers to learning *disentangled* representations where natural factors of variation in the dataset are encoded into distinct latent dimensions (Bengio et al., 2013; Higgins et al., 2018); however, recent work has proposed incorporating weak supervision in this process (Locatello et al., 2019; Shu et al., 2019; Brehmer et al., 2022). As a supervised problem without OOD shifts, diversity refers to learning functions that disagree outside training points. Methods in this space have generally made use of input gradients (Ross et al., 2017; 2018) and orthogonality (Mashhadi et al., 2021; Xu et al., 2021). Finally, diversity is considered in the context of distribution shifts — either to improve robustness against adversarial attacks (Pang et al., 2019), to disambiguate between perfectly correlated signals (Lee et al., 2022), or to evade the simplicity bias by learning more complex functions (Pagliardini et al., 2022; Teney et al., 2021). Our work is most closely aligned with this last category. Unlike the approaches above, we exploit information-theoretic measures as our objective.

**Shortcut Learning and Spurious Correlations**    Shortcut learning (Geirhos et al., 2020) involves distribution shifts arising from spurious correlations (Buolamwini & Gebru, 2018; Xiao et al., 2020; Moayeri et al., 2022) and neural network biases (architectural or simplicity biases) (Geirhos et al., 2018; Shah et al., 2020; Teney et al., 2021). Methods that tackle distribution shifts must use additional data and/or assumptions. Examples of additional data include having multiple training environments (Arjovsky et al., 2019), counterfactual examples (Teney et al., 2020), access to enough validation data to fine-tune the model (Kirichenko et al., 2022), or group labels (Sagawa et al., 2019; Puli et al., 2022). Examples of additional assumptions include exploiting the lottery ticket hypothesis (Zhang et al., 2021) or treating misclassified training examples by an initial model as a proxy for minority groups (Liu et al., 2021; Zhang et al., 2022). Unlike these methods, we aim to learn *all* predictive signals in the dataset, rather than performing well on a single test distribution. Furthermore, we use validation data for hyperparameter tuning only, without additional sources of data (e.g. group labels).

**Information Bottleneck and Conditional Independence**    The line of work most similar to ours also exploits the information bottleneck method to learn diversity. Sinha et al. (2020) minimizes the mutual information $\mathcal{I}(\hat{z}_s, \hat{z}_t)$ between learnt representations $\hat{z}_m$, however, this term is *unconditional* and will simply learn weak (biased) predictors, as noted in Section 3. Rame & Cord (2021) introduce DICE, which minimizes the *conditional* term $\mathcal{CMI}(\hat{z}_s, \hat{z}_t)$. Pace et al. (2020) considers *total correlation* $\mathcal{TC}(\hat{z}_1, \ldots, \hat{z}_M)$ instead of pairwise terms. Unlike CoDE, both of these approaches compute mutual information terms on the high-dimensional representations $\hat{z}_m$. Their objectives are intractable and must be approximated. For example, DICE requires *both* variational approximations and a jointly trained adversarial discriminator that learns to distinguish pairwise classifiers. Compared to these approaches, CoDE is by far computationally advantageous as mutual information for categorical output distributions can be computed faster *and* exactly.

## 5    EXPERIMENTS

Section 5.1 presents experiments on `ColoredMNIST`, which is used both to demonstrate the viability of our approach and to highlight pivotal observations and ablations. Section 5.2 then evaluates CoDE on larger benchmark datasets for shortcut learning to show that it scales effectively.

### 5.1    COLOREDMNIST

**Setup**    As described in Section 2, the original MNIST (LeCun et al., 1998) labels are binarized (0-4, 5-9) and used to generate true labels $y$ with noise $p_d$. $y$ then generates binary color labels with noise $p_c$, used to color the image (red or green). As per Arjovsky et al. (2019), we consider two test environments: the *training distribution* where $p_d = 0.25$ and $p_c = 0.1$, and the *adversarial distribution* where $p_d = 0.25$ but $p_c = 0.9$ (hence the shortcut-label correlation is reversed).

**Evaluation Baselines and Metrics**    As is standard in existing work, we evaluate predictive accuracy on the training and adversarial distributions. In choosing baselines, we considered the following desiderata for fairness and comprehensiveness: **(i)** comparing to both ensembling and non-ensembling methods, **(ii)** amongst ensembling methods, comparing to both conditional independence-based methods and those that do not, and **(iii)** comparing only to methods that *do not* require additional sources of data besides validation data for hyperparameter tuning. We chose the following baselines:

| $(p_d, p_c)$ | Results on ColoredMNIST | | | |
| --- | --- | --- | --- | --- |
| | **Training** (0.25, 0.1) | **Adversarial** (0.25, 0.9) | **Random-Color** (0.25, 0.5) | **Random-Color + Perfect-Digit** (0.0, 0.5) |
| Invariant | 75 | 75 | 75 | 100 |
| ERM | 88.6 | 15.3 | 52.5 | 53.4 |
| JTT | 17.8 | 87.9 | 52.5 | 56.6 |
| Ortho-Ensemble | 89.8 | 11.1 | 50.3 | 49.2 |
| TC-Ensemble | 89.1 | 69.8* | - | - |
| CoDE | 70.7 | **70.0** | 70.8 | 91.2 |

Table 1: Results on `ColoredMNIST`. A theoretically ideal classifier relying only on digit (denoted as "Invariant") will be upper-bounded by the digit-label noise $p_d$ (75%), hence any result above 75% is relying on the color shortcut. CoDE has the strongest performance on the adversarial distribution. *We were unable to reproduce TC-Ensemble on `ColoredMNIST`, and are citing their results in lieu.*

1. **ERM classifier** (ERM): single, standard classifier trained with ERM

2. **Just Train Twice** (Liu et al., 2021) (JTT): an initial classifier is trained for a limited number of epochs; mis-classified examples are upweighted to train the final classifier

3. **Ensembles using input gradient orthogonality** (Teney et al., 2021) (Ortho-Ensemble): an ensemble where the regularizing term is the dot product of the two models' input gradients

4. **Ensembles using conditional total correlation (CTC)** (Pace et al., 2020) (TC-Ensemble): an ensemble learnt by minimizing CTC over the encoder network's representation

Table 1 shows all results on `ColoredMNIST`. We discuss the most important findings below.

### 1. Enforcing conditional independence on output distributions achieves diversity effectively.

Since `ColoredMNIST` is an artificially-created dataset whose DGP we know satisfy latent conditional independence ($p_c$ and $p_d$ are independently determined), it is the ideal dataset to evaluate our key claim. Indeed, the strong performance of CoDE shows that it is sufficient to enforce conditional independence on output distributions. The final predictor selected via fast adaptation achieves near-invariant results, suggesting that it has correctly learnt from digit rather than color.

### 2. CoDE generalizes to multiple OOD test distributions, without overfitting on any one specific distribution.

In Table 1, JTT achieved about 90% on the adversarial distribution, implying that it *overfitted* to the adversarial distribution — by learning the *opposite* shortcut (color) correlation rather than the true signal (digit). This is further confirmed with additional results on two other test environments (Random-Color and Random-Color + Perfect-Digit) where $p_c = 0.5$. JTT is close to random on these two environments, suggesting that it is still relying on color as the predictive feature. In contrast, CoDE achieves 91% when $p_d = 0.0$, suggesting that it has learnt to predict using digit.

While concerning, these results are not entirely surprising. A method like JTT did exactly what it was designed to do, which is to minimize classification errors on the adversarial test distribution. Since $p_c = 0.1$, the opposite color correlation is precisely this loss-minimizing function. In contrast, CoDE will not find such a solution because two classifiers that return opposite predictions using the same feature (color) are *perfectly* correlated, even when conditioned on $y$.

These results highlight the shortcomings of single classifier methods like JTT. Such methods are designed to generalize to a specific test distribution, in general, this **does not imply** that they have learnt the desired predictive signal — merely that they have learnt an arbitrary function that does well on the test distribution. In contrast, methods that enforce diversity, such as CoDE, explicitly recover meaningful predictive signals that can generalize to *any* test distribution where $p(\mathbf{z}|y)$ changes.

|  | ColoredMNIST | | CelebA | |
|---|---|---|---|---|
|  | **Training** | **Adversarial** | **Ave** | **Worst** |
| CoDE (sequential $f_1$) | 90.0 | 10.2 | 95.2 | 31.1 |
| CoDE (sequential $f_2$) | 70.1 | 70.0 | 95.0 | 33.3 |
| CoDE (sequential $f_3$) | 63.2 | 49.0 | | |
| CoDE (sequential $f_5$) | 64.4 | 42.2 | | |
| CoDE (joint $M = 2$) | 73.4 | 60.2 | 89.2 | 83.3 |
| CoDE (joint $M = 3$) | 74.6 | 44.3 | | |
| CoDE (joint $M = 5$) | 71.9 | 43.1 | | |

Table 2: Additional results on `ColoredMNIST` and `CelebA`.

**3. Joint and sequential optimization are suited to different datasets.**

From our experiments, we found that there is no clear preference between either choice in terms of generalization ability. Table 2 shows both joint and sequential results on the `ColoredMNIST` and `CelebA` datasets. For `ColoredMNIST`, we found that sequential training performed better than joint training. For `CelebA`, joint training yielded a stronger classifier.

This might be explained by the biases of the ERM model. In `ColoredMNIST`, as both latent factors (color and digit) are noisy predictors and as color presents a particularly simple shortcut, the ERM model solely learns from color. As such, a second classifier model that is trained sequentially can learn to predict solely from the digit feature. In contrast, the ERM model in `CelebA` has likely picked up some combination of the spurious (gender) and true (hair color) features, possibly because gender gives rise to complex features that are not ncessarily simpler to learn. This corroborates previous findings indicating that ERM models can learn an arbitrary combination of all predictive signals (Zhang et al., 2021; Kirichenko et al., 2022). As such, when trained sequentially, the second model fails to learn from hair color alone.

The advantages of sequential optimization are: (i) cheaper computational costs as $M$ increases, and (ii) providing a natural stopping point for training. The latter comes from the fact that we can select for $M$ by terminating the training process when the subsequent classifier is no longer predictive, which indicates that there are no further predictive factors to be learnt. In contrast, joint optimization is advantageous as it allows us to avoid the pathological sitation where earlier models learn combinations of predictive factors. As small values of $M$ work well for CoDEs, we note that the computational cost of CoDEs are not prohibitive.

## 5.2 BENCHMARK DATASETS

**Setup**    We consider the following benchmark datasets:

- `CelebA` Liu et al. (2018); Sagawa et al. (2019): A dataset of celebrity faces with various labelled attributes. We consider the benchmark task in (Sagawa et al., 2019) of predicting the binary hair color attribute (blond or not), with gender (female or male) as the spurious attribute. There are therefore four groups.

- `Waterbirds` (Wah et al., 2011; Sagawa et al., 2019): Setup described in Section 2. There are also four groups as both latent factors (background and foreground) are binary.

- `MF-Dominoes` (MNIST-FashionMNIST) (LeCun et al., 1998; Xiao et al., 2017; Shah et al., 2020; Pagliardini et al., 2022): Each input image concatenates an MNIST digit (0 or 1) with a FashionMNIST object (coat or dress). The true label is the FashionMNIST object; the simpler MNIST feature is the shortcut. The minority groups represent 5% of the data.

Table 3 shows all results on the benchmark datasets.

**4. CoDE scales well to large datasets and retains effectiveness at preventing shortcut learning.**

| | CelebA | | Waterbirds | | MF-Dominoes | |
| --- | --- | --- | --- | --- | --- | --- |
| **Method** | Ave | Worst | Ave | Worst | Ave | Worst |
| ERM | 94.8 | 46.7 | 90.4 | 78.3 | 88.9 | 76.9 |
| JTT | 88.0* | 81.1* | 93.3* | **86.7*** | 89.5 | 76.1 |
| CoDE | 89.2 | **83.3** | 91.5 | 79.4 | 92.1 | **91.4** |

Table 3: Main results on all datasets. CoDE achieves better adversarial or wrost-group accuracy than the other methods on all datasets except `Waterbirds`.
* Results from the JTT paper. We share the same model and training environment as their paper.

On `CelebA` and `MF-Dominoes`, CoDE achieves the best worst-group accuracy. Unlike the earlier `ColoredMNIST` dataset, we have no guarantees that the core assumption of latent conditional independence holds. However, the strong performance of CoDE on these datasets shows that such an assumption is generally valid and useful when scaled to more realistic datasets.

We note that CoDE performs poorly on `Waterbirds`. In our experiments, we selected $M = 2$ as the ensemble size. Even though there are no guarantees what will be the two conditionally independent classifiers that CoDE learns, in the other datasets, the results show that they do each correspond to the shortcut and true signal. This implies that in these datasets: **(a)** there are no features conditionally independent to *both* the shortcut and true signals and yet also strongly predictive of the label, and **(b)** the shortcut or true signal cannot be decomposed themselves into conditionally independent signals. Our hypothesis is that **(b)** is not true for `Waterbirds`. As the dataset is varied and contains a range of land and water backgrounds, there could be *multiple* spurious signals in the background that are somehow conditionally independent, resulting in these signals being learnt. Another possibility is that the ensemble could have learnt an imperfect or partial foreground signal.

**5. Computational effectiveness is crucial to learn diverse ensembles at scale.**

Beyond `ColoredMNIST`, we found that it was computationally prohibitive to run Ortho-Ensemble, as the size of ensembles required to work well (48 or 96) was too high. We also noted that we could not implement TC-Ensembles successfully on larger datasets, noting that the original authors do not test on datasets besides `ColoredMNIST` either. We believe that this further highlights the importance of computational efficiency in diverse ensembling.

## 6 DISCUSSION AND CONCLUSION

Appendix B discusses potential failure modes of our method.

We introduce CoDE, a method for learning an ensemble of diverse classifiers that rely on different predictive signals in the dataset. The key assumption made by CoDE conditional independence between predictive signals, which it enforces on classifiers' output distributions. We find that CoDE works well in practice when applied to shortcut learning tasks. Future work includes: **(a)** evaluating CoDEs on other applications where multiple predictive signals exist, such as fairness-related tasks where we might want to learn classifiers that do not rely on sensitive attributes, and **(b)** considering other metrics for conditional independence that might provide more fine-grained signals than output distributions (e.g. minimizing mutual information between latent representations).

ETHICS STATEMENT

**Positive Impact**    Being robust to distribution shifts, CoDE will have a positive impact when deployed to high-stakes domains, where learning shortcut signals can have harmful social consequences. One such notable example is pneumonia prediction — models trained on pneumonia labels from chest X-ray scans have been shown to learn machine-specific artifacts in the background, which is a shortcut as hospitals have differing positivity rates and use different machines (Zech et al., 2018).

**Negative Impact**    There are no notable negative impacts of using CoDE specifically, besides the general potential for all machine learning models to be abused in the wrong hands.

REPRODUCIBILITY STATEMENT

We intend to release public code with a camera-ready version of the paper.

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

## A    EXPERIMENTAL DETAILS

**Architecture and Training Details**    For `ColoredMNIST`, we use a CNN as the classifier, containing two convolutional layers and two fully-connected layers. Adam (Kingma & Ba, 2014) is used for optimization, with a learning rate of 0.001. For `CelebA`, `Waterbirds`, and `MF-Dominoes`, we use a ResNet-50 (He et al., 2016). SGD is used for optimization, with a learning rate of 0.001, momentum decay of 0.9, and weight decay of 0.001. Additionally, following previous work (e.g. Sagawa et al., 2019; Liu et al., 2021), the `Waterbirds` model is pre-trained on ImageNet (Deng et al., 2009) and includes data augmentation in the form of random horizontal flips and random resized cropping. For `CelebA` and `Waterbirds`, class reweighting is performed to ensure that there are roughly equal positive and negative labels. The random seed used for all experiments is 13.

**Hyperparameters for CoDE and Baselines**    *CoDE.* For all four datasets, we used $M = 2$ as the ensemble size, besides ablations for $M$ as detailed in Appendix B. The results in Table 3 were achieved with sequential training for `ColoredMNIST` and with joint training for the other three datasets. For `ColoredMNIST`, $\lambda_1 = 1200$ and $\lambda_2 = 10$. For `CelebA`, $\lambda_1 = 500$ and $\lambda_2 = 0.1$. For `Waterbirds`, $\lambda_1 = 500$ and $\lambda_2 = 0.1$. For `MF-Dominoes`, $\lambda_1 = 300$ and $\lambda_2 = 0.1$. *JTT.* We performed a hyperparameter sweep with $T \in \{1, 5, 10\}$ (number of epochs for initial model training) and $\alpha \in \{2, 10, 100\}$ (upweighting factor for mis-classified examples). *Orthogonal Ensembles.* All classifiers share the same feature extractor (i.e. convolutional output for `ColoredMNIST` and ResNet-50 feature representation for the other three datasets). We experimented with different values of $M$, however, values of $M$ above 16 (for `ColoredMNIST`) and above 4 (for the other three datasets) were prohitibively expensive. As such, we did not try $M = 48$ or $M = 96$ as used by Teney et al. (2021). For these smaller values of $M$ that we tried, we did not notice an improvement from the ERM model. Besides `ColoredMNIST`, we did not report these results.

## B    MODEL MIS-SPECIFICATION: POTENTIAL FAILURE MODES

The success of any method tackling distribution shifts depends on how well the assumptions made have been upheld. We discuss the potential implications when the model is mis-specified and these assumptions are no longer valid.

**Conditional Dependence**    CoDE relies on the assumption that predictive signals are conditionally independent. We using the synthetic `ColoredMNIST` dataset to generate a DGP where such an assumption does not hold true. Instead of the standard setup where color labels are generated from the true labels, we generate color labels from the *original* (binarized) MNIST labels instead, at the same noise level $p_c = 0.1$. This means that the color and digit signals are now highly correlated. Both are still predictive since the true labels themselves were generated from MNIST labels.

Table 4 shows the results of this experiment. As we expect, conditionally dependent features cannot be recovered by minimizing conditional mutual information. The ensemble either recovers *one* of the two features (when trained sequentially) or *neither*. This confirms our intuition that conditional independence must be correctly specified for CoDE to work. While these results demonstrate a failure mode of CoDE, conditional independence between predictive factors of interest does hold well in many natural image datasets, as shown in Table 3.

**Latent Mis-specification**    The size of the ensemble $M$ specifies how many predictive latent factors we believe generated the dataset. We can consider the mis-specification of $M$ in either direction: **(i)** if the true dimension of $\mathbf{z}$ is smaller than $M$, and **(ii)** if the true dimension of $\mathbf{z}$ is larger than $M$.

In case **(i)**, since the number of conditional independent components has been over-specified, whether the ensemble has been jointly or sequentially trained makes a difference. Consider the results on the `ColoredMNIST` dataset in Table 2 again. In the sequential regime, the first two classifiers $f_1$ and $f_2$ correspond to the color and digit classifiers respectively, however, the subsequent few classifiers ($f_3$ and $f_5$) do not learn anything meaningful and perform poorly on both training and adversarial distributions. However, as noted in Section 5, this does not pose a serious problem since we can use validation data to naturally determine the stopping point. On the other hand, over-specification of $M$ is more worrying in the joint regime, as there is no guarantee that any of the true latent factors are learnt at all. As Table 2 shows, for $M = 3$ or 5, the *best-performing* classifier does not generalize.

| $(p_d, p_c)$ | Training
(0.25, 0.1) | Adversarial
(0.25, 0.9) | Random-Color
(0.25, 0.5) | Perfect-Digit
(0.0, 0.5) |
|---|---|---|---|---|
| CoDE (sequential $f_1$) | 77.1 | 63.2 | 70.6 | 90.5 |
| CoDE (sequential $f_2$) | 53.6 | 50.1 | 51.5 | 54.5 |
| CoDE (joint $f_1$) | 84.9 | 25.8 | 56.0 | 60.4 |
| CoDE (joint $f_2$) | 54.3 | 73.0 | 64.4 | 77.5 |

Table 4: Results on ColoredMNIST with color-digit conditional dependence, on both joint and sequential training with $M = 2$ classifiers. When trained sequentially, the first classifier $f_1$ learns the digit correlation since digit is most predictive in this setup. However, as color is no longer conditionally independent of digit, there is no predictive feature that can be learnt by the second classifier $f_2$, resulting in a close-to-random predictor. When trained jointly, neither of the classifiers correspond to the color or digit feature.

In case **(ii)**, where the number of conditional independent components is under-specified, the learnt ensemble may correspond to any subset of the true latent factors and individual classifiers could also learn arbitrary combinations of the latent factors. For example, the trivial case where $M = 1$ is underspecified simply returns the ERM model. In general, since $M$ is a hyperparameter, latent mis-specification does not pose a serious problem as we can tune its value using the validation data.

