# OpenReview forum: "Doing Fast Adaptation Fast: Conditionally Independent Deep Ensembles for Distribution Shifts"
_ICLR.cc/2023/Conference — Submitted to ICLR 2023_

### Official Review · Reviewer_iHMc · 2022-10-26

**Confidence:** 3
**Clarity, Quality, Novelty And Reproducibility:** The clarity needs to be further impro…
**Correctness:** 3
**Technical Novelty And Significance:** 2
**Empirical Novelty And Significance:** 2
**Recommendation:** 6

**Strength And Weaknesses:**

Pros:
* The proposed fast adaption method for ensemble learning can addresses distribution shift and shortcut learning simultaneously. These two issues are common and realistic in practice.

Cons:
* Based on the description in this paper, I find the separate predictive signals are very similar to multi-view learning. Both of them are to use the different perspectives of a dataset to perform learning tasks. More discussions about the similarities and differences between these two settings are required. If the methods of multi-view learning could be extended to this problem.

* The motivations are not clear. For example, they claim that considering the independence of latent factors conditioned on labels is better. However, I cannot get its advantages over unconditioned independence from this paper. "Dimensions of z can be dependent by virtue of their correlation to y" may be a reason. However, this explanation is too abstract, and more high-level analyses and practical examples will be better. As a way to realize the conditional independence, why do they choose the conditional mutual information? What are its advantages? If other metrics that are related to independence would make the method fail?

* The experiment design is problematic. The title of this paper emphasize the "distribution shift". However, in experiments, they mainly verify the effectiveness for shortcut learning,  and the distribution shift is not verified specifically. They should perform additional experiments on the datasets of domain adaptation or OOD generalization. Besides, due to the similarity with multi-view learning, comparing CoDE with the corresponding methods is necessary.


**Summary Of The Paper:**

This paper aims to learn a set of classifiers to take care of different predictive signals in the dataset. They propose that the "diversity" of the ensemble is important and the conditional independence is an effective way to realize this goal.

**Summary Of The Review:**

Please refer to Strength And Weaknesses.

---

> ### Author Response · Authors · 2022-11-18
> **Response to Reviewer iHMc (Part 1)**
>
> We are grateful for the time you took to review our work. As you mentioned in your review, shortcut learning is a common and realistic problem that we encounter in many real-life domains and our approach effectively mitigates shortcut learning by explicitly learning diverse factors in the dataset.
>
> Below, we address the three areas of concern that you raised in your review. After reading your comments carefully, **we believe that we are in a strong position to clarify and alleviate these concerns**, and we hope that you can engage in our response carefully.
>
> —-—-—-—-—-—-—-—-—-—-—-—-—-—-—-—-—-—-—-—-—-—-—-
>
> **(A) Relationship to multi-view learning**
>
> First of all, we emphasize that we find multi-view learning a fascinating area of machine learning that solves an important real-world problem, which is to learn better in situations where we have multiple types (“views”) of data available to us. However, even though shortcut learning and multi-view learning are tangentially related by the presence of separate predictive signals, the task setup is different and **multi-view learning methods cannot simply be extended** for shortcut learning problems.
>
> The fundamental difference here is that the multiple predictive signals in the datasets that we consider **cannot be separated apart**. In multi-view learning, the views are assumed to be independent inputs and we can train separate individual classifiers on each of the views. However, shortcut learning deals with the case where a single input contains multiple signals.
>
> For example, in the coloredMNIST dataset, it is not possible to split the input MNIST image into its digit and color components and thus treat them as separate views, without prior knowledge that color is the shortcut that we want to remove. For some datasets, even with prior knowledge, it is not clear how we can separate the signals. E.g. consider the CelebA facial dataset in [4], where the true signal is “attractive or not” and the shortcut is “smiling or not”. Here, it is not even clear how we can extract “attractiveness” as a distinct feature from the image.
>
> This inability to split up the predictive signals is precisely why shortcut learning is a challenging problem! Otherwise, we could have simply trained a classifier only on the subset of views that will still be predictive in the test distribution.
>
> This difference matters because it means that we cannot simply take multi-view learning methods and directly apply them to shortcut learning problems. Having distinct views is central to how approaches for multi-view learning work. In methods like co-training or co-regularization, we train separate classifiers on each of the views and then align the predictions of these classifiers. E.g. in co-training, the most confident predictions of a classifier trained on View A is used to generate additional training labels for a classifier trained on View B, and vice versa. This will not be possible for shortcut learning because it is not possible to train a classifier only on a certain predictive signal (while ignoring the rest).
>
> As such, we do not believe that multi-view learning methods at present are an appropriate baseline for our work. Nevertheless, we can see where the similarities are, and we hope that approaches in multi-view learning may inspire or inform even better solutions for shortcut learning in the future. **We would also be happy to include multi-view learning works as related work, in the hopes of inspiring such future work.**

---

> ### Author Response · Authors · 2022-11-18
> **Response to Reviewer iHMc (Part 2)**
>
> **(B) Shortcut learning vs. distribution shift**
>
> In your review, you raised the issue that our paper emphasizes solving distribution shifts but only performs experiments related to shortcut learning. We believe that there may be a misunderstanding here. **Shortcut learning is a distribution shift problem.**
>
> In general, we note that there are many kinds of distribution shifts and the topic of OOD generalization is a broad one. For example, domain adaptation [1], domain generalization [2], transfer learning, spurious correlations [3], etc. all refer to distinct distribution shift problems where the task and assumptions made are set up differently.
>
> Shortcut learning is one such problem. It describes the setup where the data consists of multiple features (e.g. color and shape, or image background and foreground), with the distribution shift arising from the fact that all of these features are predictive in the training distribution, however, _only a subset of these features_ are predictive in the test distribution. For example, in ColoredMNIST, both color and digit are strongly correlated to the training label, whereas at test time, we flip the color of each image and so color is no longer predictive. A more realistic example would be doing object classification in new locations or settings where the image background is no longer correlated to the label — e.g. a self-driving car’s perception model trained on data from the United States can no longer rely on U.S. traffic lights (background) to detect the presence of the pedestrian (classification target) when deployed on another country.
>
> Methods that deal with distribution shifts will not work for all types of shifts. For example, methods for domain adaptation are not generally useful for spurious correlations, methods for domain generalization cannot solve transfer learning more broadly, etc. Similarly, our work is designed specifically to deal with the distribution shifts associated with shortcut learning, and not other types of shifts.
>
> However, we acknowledge that our title may have misled you into thinking that our method is designed to be an all-purpose distribution shift solution. We have to tried to highlight shortcut learning as the specific distribution shift application in the introduction (first and last paragraphs), however, **we acknowledge that we may need to make this point more clearly and we would be happy to hear any feedback on how to improve the clarity better (e.g. changing the title to be more specific).**
>
> **(C) Unconditional vs conditional independence**
>
> In your review, you mentioned that it is not clear why we consider conditional independence as the training objective over unconditional independence. This is because conditional independence is the consequence of a dataset having multiple predictive signals.
>
> First, we note that unconditional independence is not the correct objective here because the **predictive signals are not unconditionally independent**. The easiest way to see this is by considering the data-generating process here, which is:
>
> z_1 ← y → z_2
>
> Both the predictive signals z_1 and z_2 are strongly correlated to the label y and so they are strongly correlated to each other as well. Here is an example. Consider the coloredMNIST dataset in our paper, where color and digit are the two predictive signals. In the training dataset, all red-colored images have the digits 0-4, and all green images have digits 5-9. Color and digit are not independent here: knowing that the color is green makes it extremely likely that the digit is 5-9. **The mutual information between color and digit will not be zero.** As such, optimizing for unconditional independence will not be the right thing to do here as we will not recover color and digit signals, which is what we ultimately seek.
>
> Conversely, the graph above implies that conditional independence is the correct objective. _Conditional on knowing the label_, knowing the color doesn’t provide any additional information on the digit of the input image, and vice versa. Since color and digit are conditionally independent, conditional mutual information is an appropriate objective for our ensemble.
>
> —-—-—-—-—-—-—-—-—-—-—-—-—-—-—-—-—-—-—-—-—-—-—-
>
> We hope that our response has adequately addressed your concerns. If you are satisfied with our response, we hope you may consider raising your score.
>
>
>
> [1] Xingchao Peng, Ben Usman, Neela Kaushik, Judy Hoffman, Dequan Wang, Kate Saenko. (2017) VisDA: The Visual Domain Adaptation Challenge.
>
> [2] Da Li, Yongxin Yang, Yi-Zhe Song, Timothy M. Hospedales. (2017) Deeper, Broader and Artier Domain Generalization.
>
> [3] Shiori Sagawa, Pang Wei Koh, Tatsunori B. Hashimoto, Percy Liang. (2019) Distributionally Robust Neural Networks for Group Shifts: On the Importance of Regularization for Worst-Case Generalization.
>
> [4] Lee, Yoonho, Huaxiu Yao, and Chelsea Finn. (2022) Diversify and Disambiguate: Learning From Underspecified Data.

---

### Official Review · Reviewer_aogU · 2022-10-26

**Confidence:** 3
**Correctness:** 2
**Technical Novelty And Significance:** 3
**Empirical Novelty And Significance:** 1
**Recommendation:** 5

**Clarity, Quality, Novelty And Reproducibility:**

The paper is well written. The data generation process description is detailed and informative. The notation is consistent.



**Strength And Weaknesses:**

Results on the ColoredMNIST dataset (table 1) raises some questions to me (i) does the train column refer to metrics of testing on the train set? and if so, does it show a tendency of ‘competing’ methods to overfit? -(ii) The difference of performance with TC-Ensemble Pace et al. is small, is there perhaps a computational benefit in comparison? (iii) the work by Pace et al. (2020) is included in the table but Rame & Cord (2021) not – being one of the closest references it would help to see its benchmark as well, or understand why it may not be applicable.

The model has a few possible variations: being sequential/joint, which is addressed by results in table 2. However, the two proposed loss components are not further studied beyond the hyperparameter comments on the appendices.

In terms of the data generation process or the impact of enforcing variability of the ensemble learnt it is not further linked to the results i.e. (i) is it a valid assumption or (ii) does the classifiers learnt show signs of being independent.


**Summary Of The Paper:**

This paper discusses a method for learning diverse ensembles “Conditionally Independent Deep Ensembles (CoDE)” and benchmarking its performance with shortcut learning datasets (i.e. ColoredMNIST, Waterbirds). The main objective is to enforce variability of the signals picked  avoiding to rely always on the same and/or strongest signal.

Authors aim to enforce conditional independence  on the output distributions (there are good properties of doing so as stated by authors i.e. being computationally cheaper). Also authors focus on enforcing confident predictors. That takes the form of two loss components:

(i) CMI - a conditional mutual information component that is computed on pairwise predictors/networks
(ii) R - a “confident”-prediction regularization

Both are controlled by scalar factors to weight their importance.

In terms of literature counterparts, authors reference two main papers: Pace et al. (2020) and Rame & Cord (2021)

**Summary Of The Review:**

Overall, I initially recommend rejecting this submission as results are not very strong in showing performance or computational cost improvement.

---

> ### Author Response · Authors · 2022-11-18
> **Response to Reviewer aogU (Part 1)**
>
> First of all, we would like to thank you for your kind words about the quality and presentation of our paper. We have taken care and effort to ensure that our method and experimental results are technically sound and clearly presented, as you noted in your review.
>
> We would like to address each of the points or questions that you raised below. **We have added new experimental results where relevant to back up our response, so we hope that you will take the time to read through our response carefully.**
>
> —-—-—-—-—-—-—-—-—-—-—-—-—-—-—-—-—-—-—-—-—-—-—-
>
>
> _**(1)** Results on the ColoredMNIST dataset (table 1) raises some questions to me (i) does the train column refer to metrics of testing on the train set? and if so, does it show a tendency of ‘competing’ methods to overfit?_
>
> The “training” column refers to accuracy on the training _distribution_, i.e. the input points are taken from MNIST’s _test_ set but colored in a way such that the label-color correlation is identical to the training data. Hence, there is no “overfitting” here in the typical sense where methods are capturing noise/regularities specific to the training data points. As such, an accuracy of >75% on the training distribution does show that competing methods have not correctly learnt to predict from the digit feature, and is not merely an artifact of overfitting.
>
> _**(2)** The difference of performance with TC-Ensemble Pace et al. is small, is there perhaps a computational benefit in comparison?_
>
> Indeed, we can show that our method is computationally advantageous compared to the other ensembling approaches. Below, we compare all the ensembling methods on ColoredMNIST. We report runtime as a multiple of the runtime of the naive deep ensemble. As in our original paper, we use the same architecture for the individual models in the ensemble to ensure results are comparable.
>
> | Runtime on ColoredMNIST |       |
> |-------------------------|-------|
> | Naive Deep Ensemble     | 1.00  |
> | CoDE (Joint)            | 1.68  |
> | CoDE (Sequential)       | 1.88  |
> | Ortho-Ensembles         | 24.57 |
> | TC-Ensembles            | 4.80  |
>
> As we can see from the table, CoDE has the fastest runtime (excluding the naive deep ensemble itself), with the joint approach being 3 times faster than TC-Ensembles (Pace et al.) and ~15 times faster than orthogonal ensembles. Note that these values are for ColoredMNIST: we will expect the difference to be even more stark on larger datasets as we will be using larger vision models.
>
> We show results only for ColoredMNIST as some of these methods (e.g. Ortho-Ensembles and TC-Ensembles) do not scale well to larger datasets, as we noted in our original paper. Indeed, the fact that many existing diverse ensembling methods have trouble scaling to larger datasets is a key motivation behind our work.
>
> _**(3)** The work by Pace et al. (2020) is included in the table but Rame & Cord (2021) not – being one of the closest references it would help to see its benchmark as well, or understand why it may not be applicable._
>
> First, we note that Rame and Cord did not make their code available and we had difficulty replicating the success of their work, in particular, we were not able to achieve stable adversarial training on the datasets that we consider.
>
> Furthermore, we have two additional reasons for comparing to Pace et al. only:
>
> (i) Since most benchmark datasets for shortcut learning only contain two predictive signals (the true signal vs. the shortcut signal), all our experiments consider ensembles of size M=2. When M=2, (conditional) total correlation is equivalent to (conditional) mutual information. As such, Pace et al. and Rame and Cord are actually using the **same** conditional independence objective. (Of course, they are _estimating_ this quantity _differently_.)
>
> (ii) As both Pace et al. and Rame and Cord rely on adversarial training of a discriminator network, Rame and Cord’s approach does not seem to carry a significant computational advantage compared to that of Pace et al.
>
> As such, we felt that comparing to Pace et al. was a sufficient baseline. We note that we also compare to other methods, such as orthogonal ensembles and non-ensembling approaches like JTT.

---

> ### Author Response · Authors · 2022-11-18
> **Response to Reviewer aogU (Part 2)**
>
> _**(4)** The model has a few possible variations: being sequential/joint, which is addressed by results in table 2. However, the two proposed loss components are not further studied beyond the hyperparameter comments on the appendices._
>
> In light of your concern, we include additional experimental results below, where we ablate for the two terms in Equation (7) of our paper.
>
> | Acc. on ColoredMNIST               | Training Distribution | Adversarial Distribution |
> |------------------------------------|-----------------------|--------------------------|
> | CoDE with both terms               | 70.7                  | 70.0                     |
> | CoDE with only CMI term            | 56.5                  | 44.6                     |
> | CoDE with only regularization term | 90.4                  | 9.7                      |
>
> The key result here is the second row of the table above, where we learn CoDE with the CMI term but not the additional output regularization term. We can see that the ensemble does not learn diversity successfully, instead learning a seemingly random solution with accuracies ~50% on both the training and adversarial distributions.
>
> This result can be understood as such: learning **random** classifiers is a possible mode collapse when we include the CMI term without the output regularization term, because the random classifier has (conditionally) independent outputs from that of another classifier. The additional regularization term allows us to avoid this trivial solution, since it further enforces each classifier to produce confident output probabilities rather than close-to-random values.
>
> On the other hand, having the regularization term only without CMI (the key ingredient) results in us simply learning the shortcut feature on all classifiers in the ensemble, as expected.
>
>
> _**(5)** In terms of the data generation process or the impact of enforcing variability of the ensemble learnt it is not further linked to the results i.e. (i) is it a valid assumption or (ii) does the classifiers learnt show signs of being independent._
>
> Indeed, we can verify that CoDE does indeed produce conditionally independent classifiers. The table below shows the conditional mutual information over output distributions **on the test dataset** (the adversarial distribution) between the two networks in the ensemble learnt by CoDE, compared to two networks in a naive deep ensemble:
>
> | ColoredMNIST Test CMI between the two networks |        |
> |------------------------------------------------|--------|
> | CoDE                                           | 0.0026 |
> | Naive Deep Ensemble                            | 0.1864 |
>
> We can see that CMI is significantly lower in CoDE (by a factor of 70), showing that the classifiers that CoDE learns can be verified to be conditionally independent.
>
> —-—-—-—-—-—-—-—-—-—-—-—-—-—-—-—-—-—-—-—-—-—-—-
>
> We hope that our response and additional experiments have adequately addressed your concerns. If you are satisfied with our response, we hope you may consider raising your score.

---

### Official Review · Reviewer_q3rX · 2022-10-31

**Confidence:** 1
**Clarity, Quality, Novelty And Reproducibility:** The paper is clearly written and novel.
**Correctness:** 4
**Technical Novelty And Significance:** 3
**Empirical Novelty And Significance:** 3
**Recommendation:** 6

**Strength And Weaknesses:**

Strength:
- The main strength of this paper lies in the clear exposition and development of CoDE.
- The definitions of invariance and diversity specific to CoDE are novel.
- The CoDE optimization problem is new, and its solution is efficient.
- The numerical results are promising.

Weakness:
- Novelty of the CoDE objective is unclear in light of the earlier work on conditional mutual information.

**Summary Of The Paper:**

This paper quantifies a notion of diversity for deep ensembles that facilitates efficient estimation. The authors show that it is sufficient to enforce conditional independence on the output distributions of the classifiers. This leads to their main contribution concerning the regularizing metric: conditional mutual information (CMI), efficiently computed in classification problems. The authors name this approach Conditionally Independent Deep Ensembles (CoDE).  The authors evaluate CoDE on benchmark datasets for shortcut learning.

**Summary Of The Review:**

The authors' main contribution is the development of CoDE approach and the objective for efficient learning in CoDE.

---

> ### Author Response · Authors · 2022-11-18
> **Response to Reviewer q3rX**
>
> We are thankful for your positive and supportive comments. As you mentioned in your review, our method is both effective and efficient in learning a diverse deep ensemble. Our work clearly and carefully highlights the utility of CoDE on benchmark datasets in shortcut learning.
>
> The singular issue mentioned in your review is how our work is different from existing work on conditional mutual information (CMI). We note that while other works like Pace et al. (2020) and Rame and Cord (2021) also regularize CMI (or total correlation), our key insight is that it suffices for us to enforce CMI on the networks’ _output distributions_, not their hidden representations. It is this very choice that allows our method to be much more computationally efficient, without sacrificing diversity of individual networks in the ensemble.
>
> If you are satisfied with the novelty of our contribution, we hope that you can consider raising your score.

---

### Author Response · Authors · 2022-11-18
**General Response to Reviewers**

Overall, we thank our reviewers for their insightful and supportive remarks, and we appreciate their positive reception towards the relevance and effectiveness of our work.

As all our reviewers recognize, learning a diverse ensemble is useful and important when it comes to tackling distribution shift problems such as shortcut learning, and our approach effectively does so by enforcing conditional independence. Whereas existing methods try to enforce quantities like orthogonality or conditional independence on the networks’ high-dimensional representations, our key insight is that regularizing conditional mutual information on the **output distributions** of the networks is both **cheap and effective**.

Compared to existing methods, our approach, CoDE, **(i)** brings about significant computational improvements **(ii)** without sacrificing performance in learning conditionally independent features that can generalize to test-time distribution shifts. To emphasize this point, we refer all reviewers to the additional experimental results contained in our response to reviewer aogU (c.f. points 2 and 5). We believe that our work will be a valuable contribution, both to the diverse ensemble community and also to the shortcut learning/spurious correlations community.

As all three reviewers raised separate points with little overlap, we will individually respond to each reviewer below. We have taken great care and effort in drafting our responses, and we are grateful to all our reviewers for their sustained engagement in our work.

---

### Decision · Program_Chairs · 2023-01-20

**Decision:**

Reject

**Justification For Why Not Higher Score:**

Please refer to the reviews as well as the meta-review. The paper lacks on various accounts - novelty, experiments, missing related work, as well as unclear positioning.

**Justification For Why Not Lower Score:**

N/A

**Metareview: Summary, Strengths And Weaknesses:**

This paper presents a method to (1) learn a diverse deep ensemble where the diversity among the members is ensured by regularizing conditional mutual information on the output distributions of each pair of members, (2) do a fast adaptation of the best model from this ensemble, when given a small amount of training data from a different distribution (in particular, the paper considers the shortcut learning, or spurious correlation problem, as an example of distribution shift).

Although the paper's basic idea is interesting, the reviewers pointed out several issues with the paper. In particular, (1) although it's not the most critical issue, from the title the paper appears to be about distribution shift and OOD generalization; however, it only considers a very specific type of distribution shift (shortcut learning/spurious correlation), (2) experimental results being weak in terms of performance as well as computational gains, (3) conditional mutual information based regularizers used in several prior works, which undermines the novelty further (even though the paper applies it on the output distribution as opposed to on latent representations).

The authors provided a response to the issues raised by the reviewerers. However, the concerns still linger due to several reasons, some of which are listed below:

- The original submission gave a somewhat misleading impression as proposing a diverse ensemble based method to handle the general distribution shift, whereas it only tackles the shortcut learning problem. The authors clarify this in the rebuttal; however, it raises the question about the positioning of this work w.r.t. the existing work on learning deep (or diverse deep) ensembles that tackle general distribution shift as well as shortcut learning/spurious correlations.

- The paper doesn't compare with a very relevant baseline - Rame & Cord (ICLR 2021) - "DICE: Diversity in Deep Ensembles via Conditional Redundancy Adversarial Estimation". The authors mention the lack of an available implementation as the reason for not including this baseline and say that Pace et al (2020; unpublished) is an equivalent baseline. However, this is not convincing. Rame & Cord (2021), provided a more detailed and rigorous study, on various metrics, on several datasets. Even if the implementation is not available, an attempt could be made to compare with the results reported in their paper, following the same experimental settings.

- For deep ensembles, the idea of diversity regularization via the output distribution is not new. For example, please see "Diversity regularization in deep ensembles" (Shui et al, 2018). Even though they don't use conditional mutual information, they do use the outputs of pairs of ensemble members to enforce diversity. Also see "Improving Adversarial Robustness via Promoting Ensemble Diversity" (Pang et al, 2019) and "Improving robustness and calibration in ensembles with diversity regularization" (Mehrtens et al, 2022) which discuss several regularizers on the output distributions.

- Given that other prior works on diverse deep ensembles are not limited to shortcut learning and explore their usefulness for general OOD generalization, the paper should also explore the same and compare with such methods on these problems. Without such comparisons, the paper's contribution/impact will be very limited (especially because the basic idea of using conditional mutual information is not very novel in itself).

Considering the above points and the comments from the reviewers, the paper in its current form does not appear ready for acceptance. The authors are advised to consider these points to improve the work and resubmit to another venue.